# `QDarts`: A Quantum Dot Array Transition Simulator for finding charge transitions in the presence of finite tunnel couplings, non-constant charging energies and sensor dots

**Jan A. Krzywda** [1⋆,3], **Weikun Liu** [2], **Evert van Nieuwenburg** [1,3], **Oswin Krause** [2†]

**1** ⟨aQaᴸ⟩ Applied Quantum Algorithms, Universiteit Leiden, The Netherlands
**2** Dept. of Computer Science University of Copenhagen Copenhagen, Denmark
**3** Instituut-Lorentz and LIACS, Universiteit Leiden, the Netherlands

† j.a.krzywda@liacs.leidenuniv.nl ⋆ oswin.krause@di.ku.dk ,

## Abstract

We present `QDarts`, an efficient simulator for realistic charge stability diagrams of quantum dot array (QDA) devices in equilibrium states. It allows for pinpointing the location of concrete charge states and their transitions in a high-dimensional voltage space (via arbitrary two-dimensional cuts through it), and includes effects of finite tunnel coupling, non-constant charging energy and a simulation of noisy sensor dots. These features enable close matching of various experimental results in the literature, and the package hence provides a flexible tool for testing QDA experiments, as well as opening the avenue for developing new methods of device tuning.

# 1 Introduction

## 1.1 Purpose of the simulator

Processing quantum information in semiconductor-based spin qubit devices requires detailed control over charge occupation in multiple electrically-defined quantum dots [1, 2]. In this work we introduce the Quantum Dot array transition simulator (`QDarts`, [3]), a software designed for simulating realistic charge stability diagrams, i.e. displaying regions of constant charge occupation in an *arbitrary* 2D cut through high dimensional voltage space, with a focus on a realistic sensor simulation of the selected transitions.

Our simulator represents a significant advancement in modelling realistic scenarios for state-of-the-art moderate-size devices. It provides an efficient framework for simulating the charge-occupation of the device together with a tuneable model of the sensor signal. It efficiently realizes and extends the commonly used Constant Interaction Model (CIM) [2,4,5]. As an extension to the CIM, it possesses a selection of commonly observed experimental features, such as:

- Variation of dot capacitances as a function of charge occupation,

- Effects of finite, coherent tunnel coupling between the dots, and

- Realistic modelling of the sensor dot signal, that includes tuneable noise parameters.

We envision our simulator to be useful in understanding of experimental data as well as for testing control algorithms developed for quantum dot arrays, especially for charge transition detection and state identification [6,7]. We validate these capabilities by benchmarking the simulator against a selection of state-of-the-art experiments [8].

The model purposely does not include the spin-degree of freedom, allowing for faster simulations and thus, to scale this simulator to larger devices. This enables more advanced simulations of device behaviour, such as tunnel couplings and sensor dots. However, when these advanced features are not needed, the modular nature of the simulator allows purely classical simulations of the ground states of the underlying capacitance model. In this case, using our simulator is still beneficial over previous approaches as it uses efficient algorithms to speed up computations in the capacitance model.

The goal of our simulator is to be as close as possible to real device behaviour and thus, setting up the simulation requires some of the tuning steps that are also found in real devices. For this reason, the simulator also provides tools to setup an initial tuning point, including:

- Sensor tuning

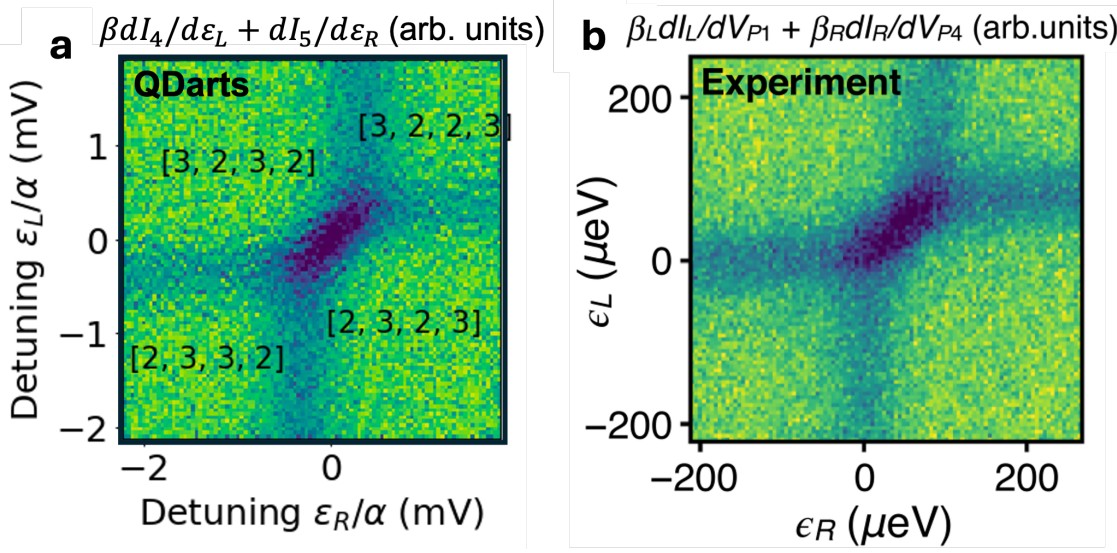

Figure 1: Reconstruction of transconductance map (gradient of conductance map from two sensor dots) as a function of detuning of two double dots. In panel **a** we show results of the `QDarts`. In **b** we have reprinted the figure with permission from S. F. Neyens et al., Physical Review Applied, 12(6) (2019). Copyright 2024 by the American Physical Society. [8]

- Virtualisation of the gates, which translates applied voltages into arbitrary combination of dots chemical potentials,

- Finding points of interests, e.g., interesting transitions in the high dimensional voltage/-chemical potential space.

## 1.2 Demonstration

To show the simulator efficiency and fidelity, we compute a 100x100 2D scan of a quadruple dot device interacting with two noisy sensor dots, which the simulator can compute on a laptop in under a minute. The outcome presented in Fig 1 **a** aims to reproduce the result of Ref. [8] which we reprinted in Fig. 1 **b**. In this example, a multi-electron transition is simulated, where two interdot transitions (horizontal and vertical lines) intersect at a point representing the simultaneous occurrence of both transitions. While [8] show that they can simulate this via a specialised model, this transition is a natural result from our simulator, obtained using a few-line prompt:

```
x, y, sensor_signal, polytopes = experiment.generate_CSD(
         plane_axes = np.array([[0,0,0,0,-1,1],[0,1,-1,0,0,0]]),
         target_state = [5,3,2,5,3,2],
         target_transition = [0,-1,1,0,-1,1],
         x_voltages=np.linspace(-0.001, 0.0008, 250),
         y_voltages=np.linspace(-0.001, 0.0008, 250),
         compute_polytopes = True,
         compensate_sensors = True,
         use_virtual_gates = True,
         use_sensor_dots = True)
```

Code 1: Code used to simulate conductance of two sensor dots sensitive to a four-dot transition from [8]

the details of which are explained in the remainder of the paper. The structure of `QDarts` and the high-level interface used to define the `experiment` are described in Sec. 2. The

arguments used in the function `generate_CSD` are sequentially introduced in Sec. 3. Finally, Sec. 4 provides a detailed explanation of the methods used in the modelling. Additionally the input data, used to model [8], and the code for generating the figures is provided in the attached Jupyter notebook.

## 2 Structure

The simulator comes with a low-level and high-level interface. The high level interface allows for a quick setup and plotting of the simulation, while the low-level interface provides fine-grained access to the full feature set. All features presented here are fully described and reproducible via the Jupyter notebooks in the `examples` folder.

### 2.1 High-Level Interface

For brevity, in the following we will only refer to the high-level interface, which is defined within the `Experiment` class. Its relation with other classes is depicted in Fig. 2. The main idea behind it, is to allow the user to easily set up instances of three main classes:

- `CapacitiveDeviceSimulator` for the electrostatic simulations,

- `ApproximateTunnelingSimulator` for effects of finite temperature and tunnel coupling between the dots

- `NoiseSensorSimulator` for the simulation of the noisy sensor dots,

using simple config dictionaries. The `Experiment` also provides a function:

- `generate_CSD` for generation of charge stability diagram and sensor signal.

Despite the transparency of the high-level interface, the fields of the config dictionaries and the arguments of `generate_CSD`, allow for accessing most functionalities of `QDarts`.

The simulator uses a multi-step approximation process. At the lowest level lies a simple ground-state simulation of some capacitance model in which charge-transitions have linear boundaries in voltage state (see method section for details). We use this simulation to inform the higher level simulation steps to compute Gibbs ensembles of states and tunnel effects of charge transitions. On top of this, we use a sensor simulation that transforms these higher level simulations into realistic sensor readout, including noise.

### 2.2 Device Representation

At the current version, the high-level interface of `QDarts` associates the number of dots with the number of plunger gates, such that for a system of six dots, the voltage space is six-dimensional. The influence of the gate-voltage on the particular dot is given by the $C_{DG}$ capacitance matrix, which for instance means that $C_{DG}[0,3]$ gives the capacitance between the dot of index 0 and the gate with index 3. The cross-capacitances between the dots are given by the $C_{DD}$ capacitance matrix. Similarly one has to specify the matrix of the tunnel couplings $t_c$ (`tunnel_couplings`).

In the high-level interface, the physical device is represented by the arguments of configuration fields. As an illustrative example, which will be used in the examples shown in Sec. 3, we model the system from Ref. [8]. It consists of six dots, two of which serve as the sensor dots, while the remaining four are occupied by individual electrons (inner-dots). The system is illustrated schematically in Fig. 3.

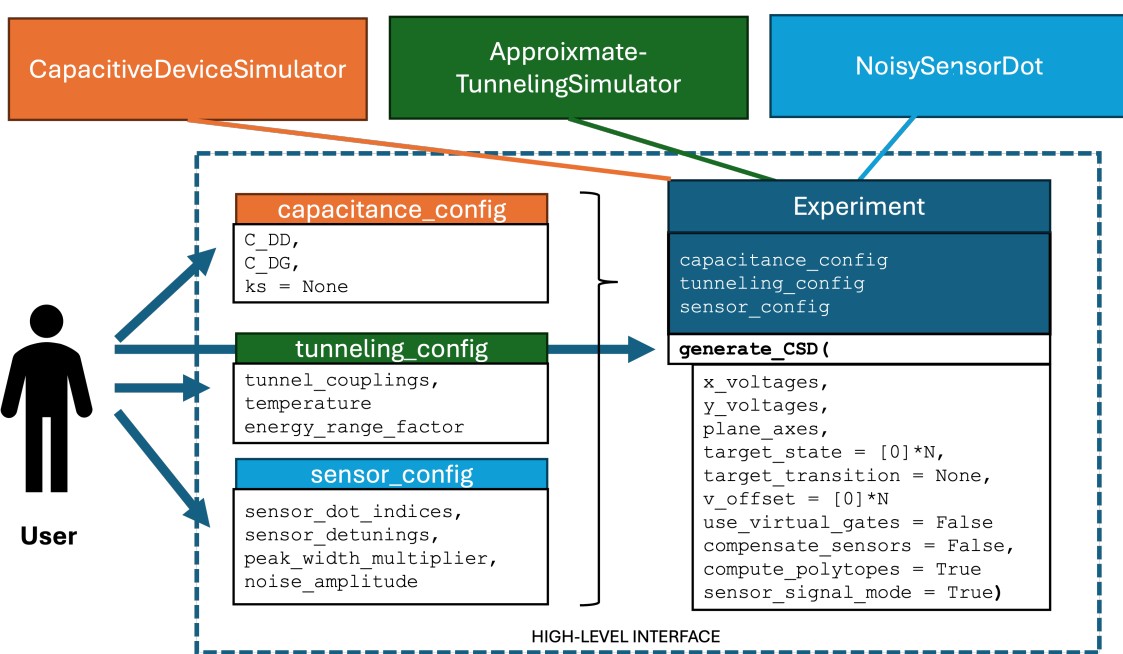

Figure 2: Class diagram illustrating capabilities of the high-level interface, explained in Sec. 2. The `Experiment` object is defined by specifying three configuration files of the form described in Sec. 3, and has a function `generate_CSD` responsible for generation of purely capacitive charge stability diagram (`sensor_signal_mode = False`) or the conductance signal from the sensor dots (`sensor_signal_mode = True`)

## 3 Features and examples

In this section we use the device to generate examples, providing a quick overview over simulator functionalities, that eventually lead to generation of Fig. 1

### 3.1 (Non-)constant Interaction Model with `CapacitanceDeviceSimulator`

As most basic functionality, `QDarts` has an ability to simulate the Constant Interaction Model (CIM) [4, 5]. The CIM is often used to interpret experimentally measured charge stability diagrams, which are obtained as 2D cuts through voltage space, using sweeps of two or more plunger gates. Within the CIM, the shape of obtained regions of constant charge occupation is set by the dot-gate capacitance matrix $C_{DG}$ and dot-dot capacitance matrix $C_{DD}$, which serve as an input to the simulator.

In our framework, the `CapacitiveDeviceSimulator` implements a purely electrostatic device simulator that represents the backbone of the later simulation steps. This simulator governs which charge configurations exist and which are neighbours of each other. Underlying the simulation is an energy function that is linear in the plunger gate voltages and the simulator uses it to compute a ground-state (See Sec. 4.1 for the details of capacitance model used).

The package includes an extension of the CIM that aligns it with the experimental observation that charging energies are a function of dot charge occupation, as a result of increase in dot size, which decreases energy cost needed to add another charge [9]. An extension of CIM is implemented by adding an additional parameter, `ks`, that results in the power law $\sim 1/N$ decay od Coulomb diamond size, according to phenomenological formula derived in Eq. (3). For illustration, in Fig. 4 **a** we show a CSD with $ks = 4$, and in panel **b** the cor-

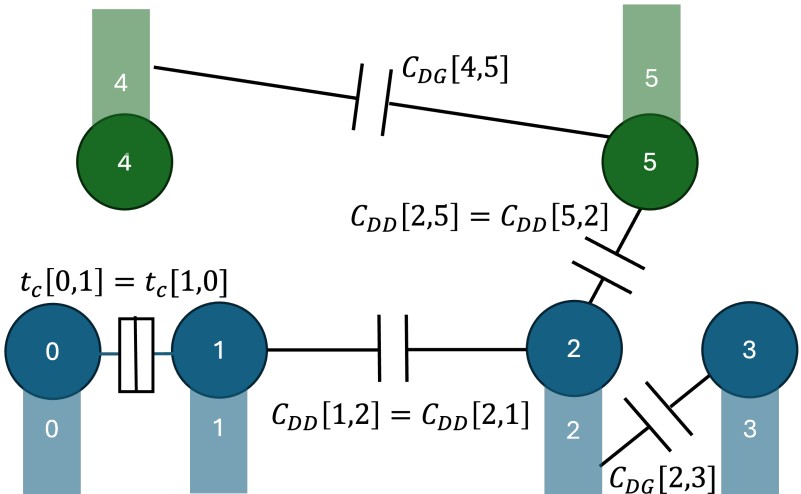

Figure 3: Illustration of a device representation inside the QDarts. The user has to specify the dot-dot capacitance matrix $C_{DD}$ (C_DD), the dot-gate capacitance matrix $C_{DG}$ (C_DG) as well as the tunnel coupling matrix $t_c$ (tunneling_matrix). Size of the arrays corrsponds to number of total gates, here six. Inside the configuration file, the user has to specify which dots are the sensors, here sensor_dot_indicies = [4,5] For the values of the parameters used in Fig. 1 see attached Jupyter notebook with examples.

responding width of the Coulomb diamond $\Delta v_n$ in relation to region with one charge $\Delta v_1$. At the current version of QDarts, the result does not include single peaks of the charging energies due to shell-filling, however still quantitatively reconstructs the initial trend visible in the experimental results [9] for a small number of electrons.

Within the high-level interface, the relevant parameters of CapacitiveDeviceSimulator are set by the configuration file of the form:

```
capacitance_config = {
        "C_DD" : C_DD,
        "C_DG" : C_DG,
        "ks" : ks}

capacitance_experiment = Experiment(capacitance_config)
```
Code 2: Configuration file of the CapacitiveDeviceSimulator object

in which skipping the last argument ks will result in the regular model with constant $\Delta v_n = \Delta v$.

## 3.2 Generating charge stability diagrams

The experiment defined above contains the function generate_CSD, which is responsible for computing electrostatic charge stability diagrams and sensor scans (See Sec. 3.5 for the latter). Within its arguments, the user has to define the orientation, origin, resolution and the size of the cut in multidimensional voltage space. This is done by the following code:

```
x,y,csd_data, polytopes = capacitance_experiment.generate_CSD(
                x_voltages = np.linspace(0, 0.07, 300),
                y_voltages = np.linspace(0, 0.03, 150),
                plane_axes = [[1,0,0,0,0,0],[0,1,0,0,0,0]],
                compute_polytopes = True)
```
Code 3: Example call of generate_CSD for computing a CSD

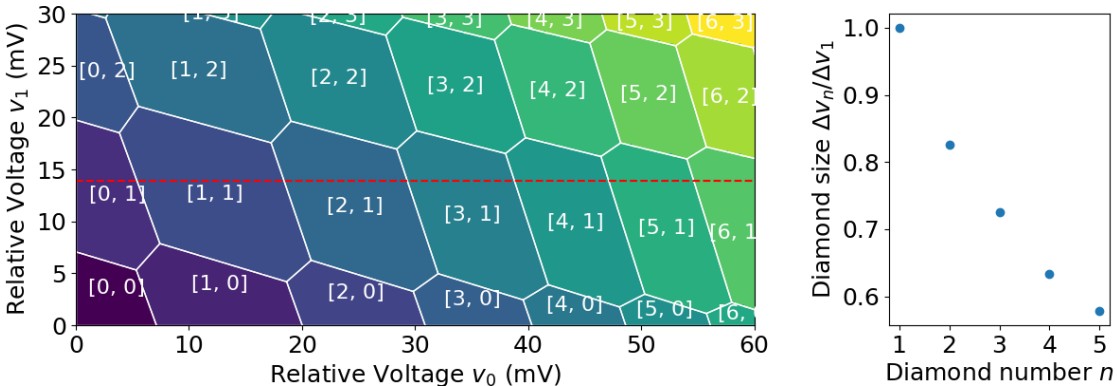

Figure 4: **a** Charge stability diagram for two-dot subsystem generated using non-constant interaction model. **b** The cut through dashed line in panel a, illustrating relative size of Coulomb diamond $\Delta v_n / \Delta v_1$ as a function of electrons that occupies dot 0.

in which, `x_voltages` and `y_voltages` are the list of numbers providing the range and resolution of the CSD, while `plane_axes` gives the vectors in voltage space, which span the cut. Additionally one can define the non-zero `v_offset` to apply a constant bias to the gates, i.e. define the new origin of voltages space around an interesting region of voltage space. However more efficient tuning is implemented, using transition selection mechanism described in Sec 3.3. In absence of gate virtualisation the vectors are defined as combination of voltages, while with virtualisation the axes become aligned with a given transition (See next Sec. 3.4). As a result in the example from Code 3, the x- and y-axes correspond to $v_0$ and $v_1$, respectively. The code returns the x and y voltage coordinates, the `csd_data` - an array representing regions of constant charges inside the space given by x and y, and `polytopes` - a dictionary containing the labels of the regions and their corners. These variables are used to generate the charge stability diagram shown in Fig. 4.

## 3.3 Transition selection

In many practical applications, the analysis of charge stability diagrams is performed in terms of relative voltages, measured with respect to a particular point of interest, `v_offset`. For instance, in the double dot one often concentrates on the vicinity of the transition that moves an electron between the dots. The package contains tools to compute such an offset automatically. Within the high-level interface, this can be done by specifying a pair of state (`target_state`) and a transition (`target_transition`) from it, which is then used to associate the origin of the voltage space with the middle of the transition. For example, to specify the transition $[1, 1] \to [0, 2]$ in a double dot, we could set `target_state = [1,1]` and `target_transition = [-1,1]`. Two more complex examples are shown in Fig. 5.

## 3.4 Gate virtualisation and chemical potential

In a realistic system, changes in voltage of one gate often affects the chemical potential of multiple quantum dots (referred to as cross-talk) [10]. These effects are due to off-diagonal elements of the capacitance matrices. It is convenient to define virtual gates, the linear combinations of plunger voltages $u_i = \sum_j \alpha_{ij} v_j$, with parameters chosen such that a change of $u_i$ only affects the chemical potential of the $i$th dot [11, 12]. In this coordinate system the transitions that add a single electron to a dot are perpendicular to each other, and the normals of the transitions that add an electron to dot $i$ are aligned with the $i$th coordinate axis. QDarts

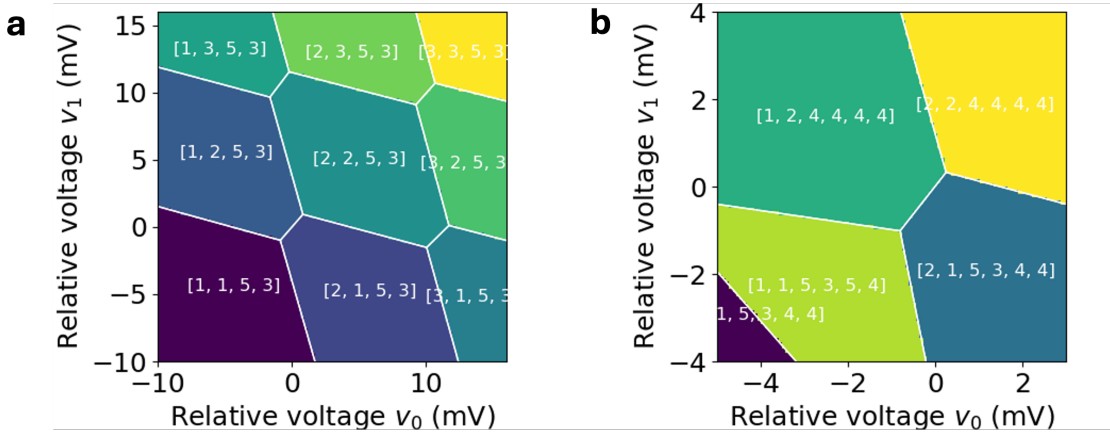

Figure 5: Illustration of automatic tuning to a selected transition point. a) Two-dot transition generated with `target_state= [2,1,5,3,4,4]`, `target_transition = [-1,1,0,0,0,0]` arguments of the `generate_CSD`, b) Four-dot transition generated with `target_state= [2,1,5,3,4,4]`, `target_transition = [-1,1,-1,1,0,0]`.

implements this feature in an extended way, where instead of being limited to transitions that add a single electron, the user is allowed to select arbitrary transitions to derive the orthogonal basis from.

This functionality is activated by adding `use_virtual_gates = True` as an argument of `generate_CSD`. The user decides which set of transitions is supposed to be orthogonal to each other and which plunger gate axes they are parallel to. With virtual gates, this is done by overloading the argument `plane_axes`. For instance, to generate Fig. 1 using Code 1, we have used `plane_axes = [[-1,1,0,0,0,0],[0,0,1,-1,0,0]]` to align the x-axis with the transitions that removes 1 charge from dot 0 and adds 1 to dot 1, i.e. with the charge transition from dot 0 to dot 1. Analogously, the y-axis would be associated with transfer from dot 3 to dot 2.

A simple example of CSD with virtualized gates is depicted in Fig. 6. We concentrate there on a two-dot subspace that, shows relatively high cross-talk. It manifest itself as skewed CSD in panel **a**. After applying `use_virtual_gates = True`, the transitions become parallel to the selected axes.

## 3.5 Sensor model with `NoisySensorDot`

We now introduce the model of charge sensing, which is experimentally implemented by measuring the current through the *sensor dots* capacitivly coupled to the *target dots*. The sensor dot has to be appropriately tuned to the side of so-called Coulomb peak [13], where conductance of the dot is sensitive to small variation of the electric field due to nearby charge movement.

To reflect the experimental reality, `QDarts` allows for a flexible tuning of sensor dot parameters, which includes the noise. The sensor model `NoisySensorDot` computes the sensor signal as the conductance, and has as parameters: the width of the Coulomb peaks as well as the amplitudes of low- and high-frequency noise. Noise is implemented purely as a variation in the position of the sensor peak due to charge effects on the device and not through effects of other measurement noise. In the `NoisySensorDot`, low-frequency noise is assumed to be constant within the measurement of a single sensor signal value and vary when computing between consecutive measurements (e.g., a voltage ramp or a row of a measured CSD). The low frequency noise is assumed to be generated by some stochastic process, for exam-

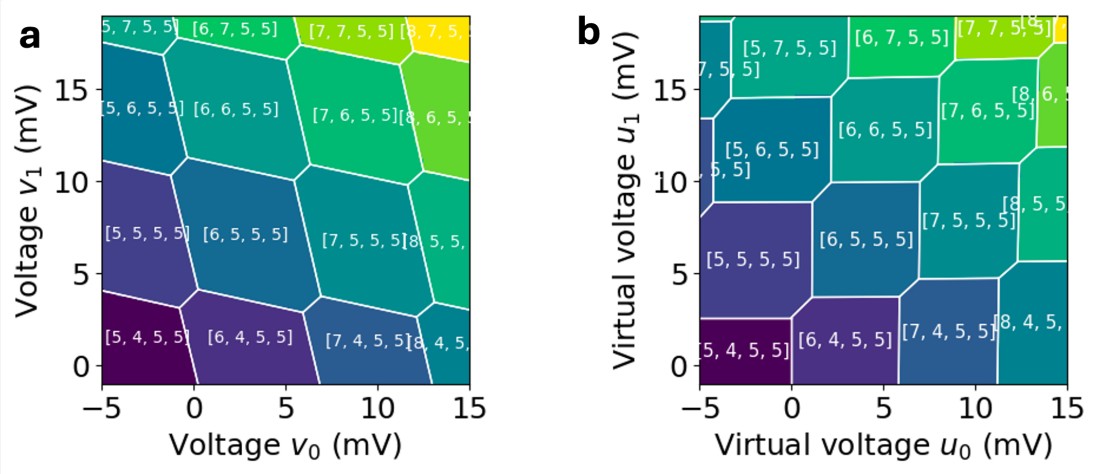

Figure 6: Alignment of the transitions, i.e. gate virtualisation. **a**) Charge stability diagram without gate virtualisation. **b**) The same but with use_virtual_gates = True argument. The axes are now aligned with the transitions, which makes them proportional to the dots chemical potentials.

ple the Ornstein-Uhlenbeck process or 1/f noise [14]. The high-frequency noise is normally distributed with a given variance and mean zero.

Within the high-level interface the properties of the sensor dots are set by the second configuration dictionary:

```
1 sensor_config = {
2     "sensor_dot_indices": [4,5],
3     "sensor_detunings": [-0.0005,-0.0005], #eV
4     "noise_amplitude": {"fast_noise": 2*1e-6, "slow_noise": 1e-8} #eV,
5     "peak_width_multiplier": m},
```

Code 4: Code for configuration of NoisySensorDot

which identifies the indices of the sensor dots (sensor_dot_indices), their detunings with respect to the Coulomb peak (sensor_detunings), the widening factor of the Coulomb peak with respect to a thermal width $m \times k_B T$ (peak_width_multiplier) and the amplitudes in fast and slow noise, given in eV (noise_amplitude). For more detailed description of the methods used in sensor dot modeling see Sec. 4.4.

The gallery of obtained charge stability diagrams (maps of sensor dot conductance), for a different noise types and varying temperature is shown in Fig. 7 **a**, while in Fig. 7 **b** we show corresponding derivatives of the conductance map, obtained from **a** using standard image processing tools. We can see that the fast noise leads to grainy images, slow noise adds streaks to the plots due to the correlated noise (with values in the 2D plot computed row-wise from left to right) and increase in temperature to overlapping state distributions.

## 3.6 Sensor scan and compensation

The scan of the sensor dot can be generated by setting use_sensor_dots = True inside the generate_CSD function, which then returns the sensor data as a matrix for each sensor in the csd_data return value.

Since the device follows the underlying capacitance model for all dots, the sensor is, as in real devices, affected by the cross-talk induced by the interdot capacitances, which modifies the position on the Coulomb peak. To prevent this from happening , i.e. to keep the

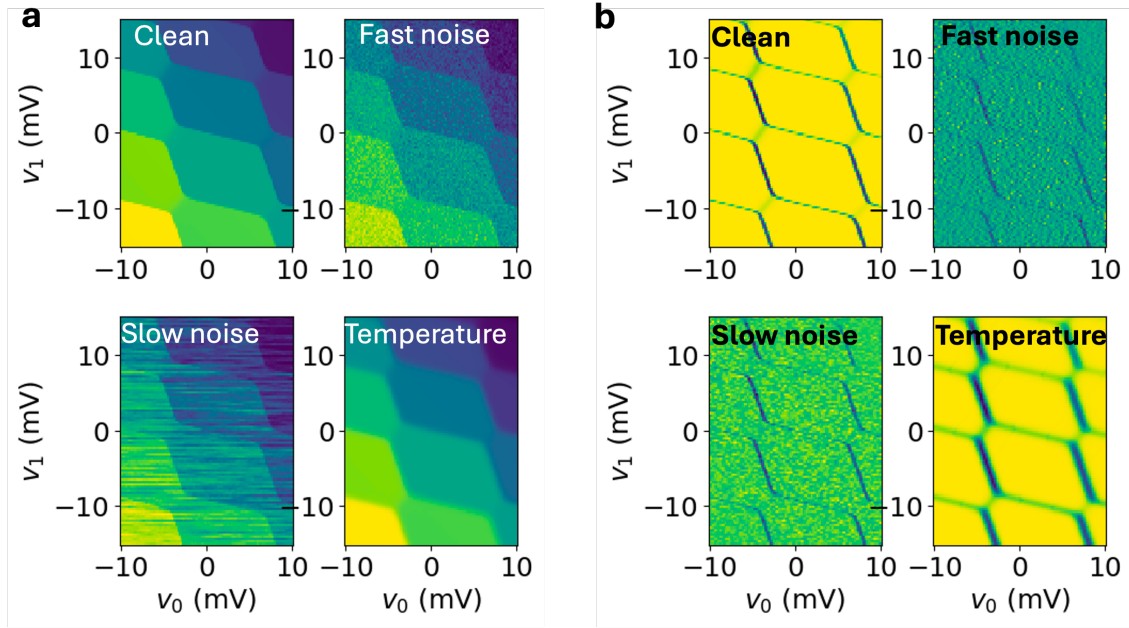

Figure 7: Gallery of different tunable noise parameters affecting performance of the sensor dot. **a** Generated conductance of the sensor dot: without noise, with fast noise, with slow noise and with increased temperature. The results are normalised to maximum conductance. **b** Numerically computed gradient of conductance along the x-axis, i.e. $\partial g / \partial v_0$.

detuning with respect to Coulomb peak constant, we need to find the compensated coordinate system and find a tuning point for the sensor. This compensation is enabled by setting `compensate_sensors = True` in `generate_CSD` (See Fig. 8 **b** for the example). The compensation is computed using the normals of the transitions at the selected `target_state`, which for this reason needs to be specified. As a result of this, the compensation of the sensor signal is affected by specifying the `ks` in the capacitance model. In this case, the compensated sensor signal will only be constant for voltage vectors inside the polytope of the selected state. Again, this is in line with experimental observations.

## 3.7 Tunnel coupling simulation with `ApproximateTunnelingSimulator`

Creating a coherent superposition of different occupation states is crucial for running many quantum algorithms. This is not possible, without tunnel coupling i.e. the element that allows for lowering the energy of the electron by coherent tunnelling events. However its presence means the occupation number is no longer a good quantum number, and the system eigenstates are coherent superposition of different charge configurations, which exponentially increases the Hilbert space [15]. However such mixing is local in energy, i.e. meaningful superposition is created between the states that are closest in electrostatic energy.

To implement this, `QDarts` uses the aforementioned capacitive model to firstly generate the uncoupled electrostatic Hamiltonian, than include tunnel coupling between the relevant dots and finally solve for the thermal states at a given `temperature` (See Sec.4.3 for more details of the model used). This takes place inside the `ApproximateTunnelingSimulator`, which provides the methods to compute the mixed state, given matrix of tunnel couplings (tunnel_couplings). Within the high-level interface this can be set up by the final configuration file:

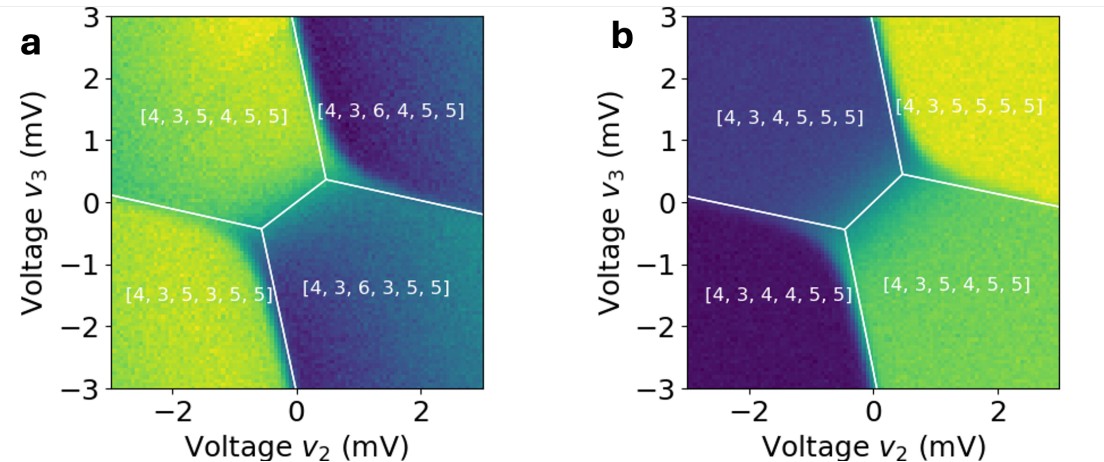

Figure 8: Simulation of the finite tunnel coupling effect. **a** Zoom into double dot transition using parameters `target_state = [4,3,6,3,5,5]` and `target_transition = [0,0,-1,1,0,0]`, with indication of non-zero tunnel coupling creating difference between the sensor signal (background and results of CIM (white lines)). **b** The same, but with compensated sensors, i.e. `compensate_sensor=True` argument of `generate_CSD`. Sensor compensation fixes the drift of the conductance within a single polytope, that can be seen in **a**.

```
1  tunneling_config = {
2         "tunnel_couplings": tunneling_matrix,   #eV
3         "temperature": 0.1, #K
4         "energy_range_factor": 20,
```
Code 5: Code for configuration of `ApproximateTunnelingSimulator`

Naive implementation of this is very expensive, so the simulator restricts the size of the Hamiltonian by only taking into accounts superpositions of charge configurations that do not have a too large energy difference (as defined by the user via `energy_range_factor`) compared to the ground state. Decreasing this number would speed up the computation, but at the cost of inaccurate results.

The resulting effect of the finite tunnel coupling can be seen in Fig. 8. Especially, note how the tunnel coupling widens the interdot transition and thus expands it compared to the ground states of the underlying capacitive model (white lines and labels).

## 4 Methods

In the following, we will describe the simulator with all its core assumptions and the modelling of each part. We assume that we are given a device with $K$ quantum dots controlled by $M$ gate voltages, $v \in \mathbb{R}^M$.

To describe the device behaviour, we assume a simplified electron model, where electrons are charges without spin. We further assume that quantum dots can be described via a set of orbitals, where each orbital can hold a single charge due to Pauli exclusion principle. We assume that these orbitals are ordered, but have the same energy level and only the electron on the outermost filled orbital is mobile. Thus, moving an electron from one quantum dot to another requires to move the electron from the outermost orbital of the first dot and filling the first unfilled orbital in the neighbouring dot. Note that even though we assume that all orbitals have the same energy level, Coulomb repulsion still affects charges on the same dot and thus

the orbitals are still separated by the energy it takes to overcome the coulomb repulsion force. As a result of these strong assumptions, the state of each dot is fully described by its location on the lattice and the number of filled orbitals (i.e., number of electrons per dot).

Based on these assumptions, we describe the Fock basis of the approximated quantum system as $|n\rangle = |n_1, \ldots, n_K\rangle$, where $n_k$ the number of electrons on dot $k$. We will now describe creation, annihilation and counting operator in this choice of basis. We define the counting operator $\eta_i |n\rangle = n_i |n\rangle$ as the number of filled orbitals. Further, we define the creation operator $c_i^\dagger$, on the $i$th dot location via its action

$$c_i^\dagger |n\rangle = |n + e_i\rangle$$

where $e_i$ is the $i$th unit vector, which can be interpreted as creating an electron on the lowest unfilled orbital. Finally, the annihilation operator $c_i$ is given by the simplified relation

$$c_i |n\rangle = |n - e_i\rangle, \quad c_i |n_1, \ldots, n_{i-1}, 0, n_{i+1}, \ldots\rangle = 0$$

We omit the change of sign in the wave function of fermions as it does not make any difference in the computed results of our simulation. For this choice of operator, it holds that $c_i$ and $c_i^\dagger$ are hermitian conjugates of each other. Given these operators, we define the Hamiltonian as

$$\mathcal{H} = \mathcal{H}_E + \sum_{i=0}^{K} \sum_{j=0}^{K} t_{ij} \left( c_i^\dagger c_j + c_j c_i^\dagger \right) \tag{1}$$

Here, $t_{ij}$ is the tunnel coupling between dots $i$ and $j$ and $\mathcal{H}_E$ is a diagonal operator where $\langle n | \mathcal{H}_E | n \rangle$ is the electrostatic energy of the electron configuration in a chosen capacitive model. Finally, let $\mathcal{N} \subset \mathbb{N}^K$ be a set of electron configurations. We will define with $\mathcal{H}_\mathcal{N}$ the projection of $\mathcal{H}$ on the subspace span$\{|n\rangle, n \in \mathcal{N}\}$.

Given the Hamiltonian, it is clear that exact simulation of the system becomes prohibitively expensive as $K$ grows. This is due to the large size of the Fock basis, which in many approaches is chosen as $n \in L^K$, where $L \in \mathbb{N}$ is the chosen maximum number of electrons. Instead, of simulating on this whole set, we will only consider a subset of states. We use as idea that given moderate tunnel coupling strength, the ground state is a linear combination of a small number of basis states of the Fock basis, while a majority of states can be discarded. Indeed, for most relevant cases, the electron configurations of the relevant states can be created from each other via a small amount of electron movements.

We will exploit this idea as follows. We will choose a simple capacitive model, with capacitive energy $E(v, n)$ in which $E(v, n) - E(v, n') \leq 0$ are linear equations in $v$. Due to this, the set of voltages that have $n$ as ground state (assuming $t_{ij} = 0$)

$$P(n) = \{v \in \mathbb{R}^M \mid n = \arg\min_{n'} E(v, n')\} \tag{2}$$

form a convex polytope, see Fig. 9. Each facet of the convex polytope fulfils $E(v, n) - E(v, n') = 0$ for some $n'$ and crossing the boundary from inside $P(n)$ leads to a state transition from state $n$ to the new state $n'$. Thus, by enumerating the facets of $P(n)$, we can derive an appropriate neighbourhood $\mathcal{N}$ and use that to obtain a low dimensional approximation $H_\mathcal{N}$ of $H$. We will extend this idea to also include states whose linear boundaries almost touch $P(n)$ in order to include states that still strongly interact with $n$ via tunnelling.

With this choice of $H_\mathcal{N}$, we can compute the mixed state matrix and use it to derive a sensor signal. The next sections describe this in more detail.

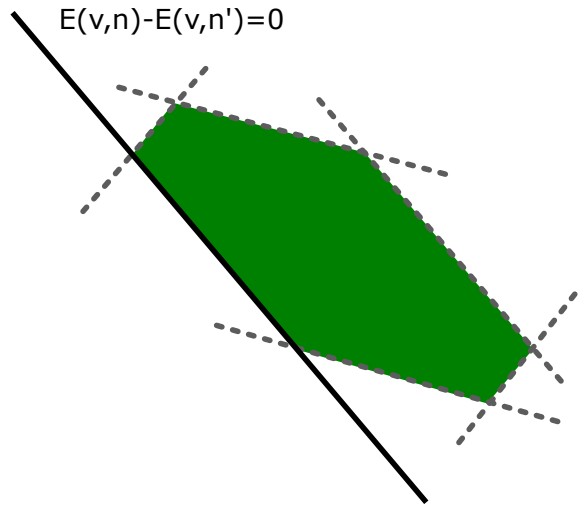

Figure 9: Visualisation of the boundaries of a ground state polytope $P(n)$. Each transition to a state $n'$ gives rise to a linear boundary. We marked with green points for which $E(v, n) - E(v, n') \leq 0$ for all $n' \neq n$

## 4.1 Constant Interaction Model

We will now describe the capacitive model, of which the most common one is the Constant Interaction Model (CIM). Following [7], this model assumes that the plunger gates and dots are purely capacitively coupled and that the relation between charge and voltage is given by the Maxwell capacitance matrix

$$
\begin{bmatrix} q_D \\ q_G \end{bmatrix} = \begin{bmatrix} C_{DD} & C_{DG} \\ \hline C_{GD} & C_{GG} \end{bmatrix} \begin{bmatrix} v_D \\ v_G \end{bmatrix} \ .
$$

Here, $C_{DD} \in \mathbb{R}^{K \times K}$ are the interdot capacitances, $C_{DG} \in \mathbb{R}^{K \times M}$ are the capacitances between dots and plunger gates. For simplicity, we assume $C_{GG} = 0$ as the value will cancel in later derivations.

In this model, the free energy is given by

$$
E(v, n) = \frac{1}{2}(n|e| - C_{DG}v)^T C_{DD}^{-1}(n|e| - C_{DG}v) \ .
$$

As a result of this model, the transition boundaries between two states are linear boundaries, as

$$
E(v, n) - E(v, n') = \frac{|e|^2}{2}(n^T C_{DD}^{-1} n - {n'}^T C_{DD}^{-1} n') - |e|(n - n')^T C_{DD}^{-1} C_{DG}v \ .
$$

Note, that the quadratic term in $v$, that is present in $E(v, n)$ cancels out in the energy difference.

## 4.2 Non-constant capacitance model

In the CIM, the charging energies of electrons remains constant, which is in contrast to experimental data where the spacing follows a power-law [9]. Moreover, in the CIM the polytopes $P(n)$, for $n_i > 0, \forall i$ are symmetric, again this is in contrast to experimental data, where transitions that add or remove an electron are not quite parallel to each other and thus the polytopes are not fully symmetric. We therefore showcase the flexibility of our simulator, by using an extension of the CIM, in which $C_{DD}$ and $C_{DG}$ are matrices that depend on $n$. We introduce a

new symmetric matrix $S(n) \in \mathbb{R}^{K \times K}$ and set $\hat{C}_{DD}(n) = S(n)C_{DD}S(n)$ and $\hat{C}_{DG}(n) = S(n)C_{DG}$. In this model, the transition boundaries are given by:

$$E(v, n) - E(v, n') = \frac{|e|^2}{2}(n^T S(n)^{-1} C_{DD}^{-1} S(n)^{-1} n - n'^T S(n')^{-1} C_{DD}^{-1} S(n')^{-1} n')$$
$$- |e|(S(n)n - S(n')n')^T C_{DD}^{-1} C_{DG} v \ .$$

Note that this equation can break the symmetry of the polytopes due to the change of the linear term via $S(n)n - S(n')n'$. Moreover, as $S(n)$ increases, the more the linear term grows compared to the constant term (in $v$) with the result that smaller changes in $v$ are needed to add additional electrons. We chose this model due to its (relative) simplicity in derivation.

We can now choose an $S(n)$ that provides the correct scaling behaviour. As computing $S(n)$ is challenging and only little is known about the scaling behaviour of interdot capacitances, we compute a diagonal $S(n)$ by considering the scaling behaviour for a device with a single dot. In this case, $S(n)$ and $C_{DD}$ are real numbers, $n$ is an integer and for simplicity we assume that we also only have a single plunger gate. With this, we can solve the previous equation for $v$ and obtain as the transition boundary between electron configurations $n$ and $n'$ the gate voltage

$$v_{n,n'} = \frac{|e|}{2C_{DG}}\left[\frac{n'}{S(n')} + \frac{n}{S(n)}\right]$$

For modelling the experimental results of [9], we are not interested in the absolute gate voltages at which a transition happens, but the relative difference of voltages between two transitions, which we require to be proportional to a power law. We pick as Ansatz:

$$\Delta v_{n+1} \equiv v_{n+1,n+2} - v_{n,n+1} = \frac{|e|}{2C_{DG}}\left[\frac{n+2}{S(n+2)} - \frac{n}{S(n)}\right] \overset{!}{=} \alpha(k)\frac{|e|}{C_{DG}}\frac{k+2}{k+n+2}. \quad (3)$$

This $\Delta v_{n+1}$ can be understood as the width (in voltage) of the polytope in the transition direction. In the equation, $\alpha(k)$ is an arbitrary proportionality factor and $k \geq 0$ is a smoothing parameter that modifies how strongly additional charges affect the capacitances. As $k \to \infty$ the CIM is obtained in the limit. We chose the right side such, that for $k > 0$ and $\alpha(k) = 1$, the difference $v_{1,2} - v_{0,1}$ is the same as in the the constant interaction model. Solving for $S(n+2)$, we obtain

$$S(n+2) = \frac{n+2}{\frac{2\alpha(k)(k+2)}{n+k+2} + \frac{n}{S(n)}}$$

This recurrence relation leaves $S(0)$ and $S(1)$ unspecified. We set $S(0) = S(1) = 1$ to keep the capacitance values for those states the same as in the CIM and we are only left with specifying $\alpha(k)$ (equivalently one could set $\alpha(k) = 1$ and use a different definition of $S(0)$). Since the recurrence relation separates $S(n)$ into two recurrence relations for even and odd $n$ the relative scaling of $S(n)$ and $S(n+1)$ is left unspecified. We choose as constraint that $S(n)$ should be linearly increasing with $n$. As we were unable to find a closed form solution, we fitted $\alpha(k)$ empirically for $1 < k < 100$ and ensured linearity in the range $n \leq 1000$. We obtained $\alpha(k) = 1 - 0.137 \cdot 3.6/(k+2.6)$.

We applied these results to the general case by setting the diagonal elements $S(n)_{ii}$ to the value that the $i$th dot would be assigned in the 1D model for its occupation $n_i$. This model works well as an extension for small electron numbers, e.g., $n_i \leq 4$ for $k = 1$. It is again important to note that the user can choose the capacitance model freely, and this extension of the CIM is only an additional choice.

## 4.3   Tunneling Simulation

After having chosen a capacitance model, we can define the full Hamiltonian (1). Still, as the number of dots grows, the size of the Hamiltonian grows quickly without bound, making any computations of ground or mixed state difficult. As outlined before, we will simplify the task by computing a neighbourhood of states. One approach is to compute the facets of the polytope $P(n^*)$ given in (2), where $n^*$ is the ground state of the capacitance model at the chosen gate voltages $v$. For any point within the polytope, this set of states is a good initial approximation. However, it is likely incomplete, even for small tunnel couplings. This is because there might be states whose transition boundaries are barely not touching $P(n)$ - often state transitions that require moving of several electrons simultaneously and which might have significant effects on the superposition of states that we aim to approximate. Thus, during computing of $P(n^*)$, we will add some slack to the computation that keeps transitions that might be relevant for tunnel coupling.

For this, let $\mathcal{N}_B(n^*)$ be a basis set of neighbours we consider as possible candidates for transitions from the state $n^*$. We pick $\mathcal{N}_B(n^*) = \{n \mid n_i \geq 0, n_i \in \{n_i^*-1, n_i^*, n_i^*+1\}, i = 1, \ldots, K\}$, the set of all states that can be reached from $n^*$ by adding or removing at most one electron on each dot. This initial set is a good approximation to the full state space for small tunnel couplings. However, it still grows exponentially with $K$. To compute the set of relevant neighbours $\mathcal{N}(n^*)$ for computation of the mixed state matrix, we compute for each transition $n' \in \mathcal{N}_B(n^*)$ the solution of the optimization problem:

$$\min_{v,\epsilon} \quad \epsilon \tag{4}$$
$$\text{s.t.} \quad v \in P(n^*)$$
$$E(v, n') - E(v, n^*) - \epsilon = 0$$
$$\epsilon \geq 0$$

The optimal solution is a point $v$ inside the polytope that minimizes $\epsilon$. If the transition from state $n^*$ to $n'$ has a facet on the surface of $P(n^*)$, then $\epsilon = 0$ as there exists a point $v' \in P(n^*)$ that fulfills $E(v', n^*) - E(v', n') = 0$. Otherwise, we have $\epsilon > 0$ in which case $\epsilon = E(v, n') - E(v, n^*)$ is the energy difference between the states. If we consider a double dot where dots are coupled by tunnel coupling $t_{1,2}$ in the Hamiltonian (1) and assuming $\epsilon \gg t_{1,2}$, then since $\epsilon$ is the minimizer of problem (4), it holds for all points $v \in P(n^*)$ that

$$E(v, n^*) - E(v, n') \geq \epsilon \gg t_{1,2} \ ,$$

and thus the state can be ignored. Thus, we can pick an upper bound $\epsilon_{\max}$ and include any state from $\mathcal{N}_B(n^*)$ in $\mathcal{N}(n^*)$ for which $\epsilon < \epsilon_{\max}$ in problem (4). Finally, for the sensor dot simulation (see next section), we need to compute energy differences of states with a different number of electrons on the sensor dot. To ensure that these pairs are available for all states of interest, we additionally allow the user to extend the region further. If a state $n$ is added to the neighbourhood, then also $n \pm k e_i$, $k = 0, \ldots, k_{\text{sens}}$ where $i$ is the index of the sensor dot $e_i$ is the $i$th unit vector and $k_{\text{sens}}$ a parameter given by the user and which is typically one or two.

With this neighbourhood we can define our approximation to the mixed state at temperature $T$ as

$$\rho(v) = \exp\left(-\frac{1}{k_B T} \mathcal{H}_{\mathcal{N}(n^*)}\right), \ n^* \text{ s.t. } v \in P(n^*) \ .$$

Here, $\exp$ is the matrix exponential. The choice of $n^*$ is unique as long as $v$ is not on a boundary of $P(n^*)$, otherwise we pick any of the possible solutions. With an appropriate choice of $\epsilon_{\max}$, this choice will not matter.

## 4.4   Sensor Simulation

We base our analysis on the influential results [16]. For our simulator, we have to adapt the result to our setting, including the effect of multiple dots and sources of electrostatic noise. For an electron configuration $n$, we will denote with $n + e_i$ ($n - e_i$) the state with one electron added (removed on the $i$th dot. Following [16], we assume that the sensor dot can directly exchange electrons with two reservoirs with energy levels that follow the Fermi-Dirac distributions. We further assume that there exists a current between the two reservoirs via the sensor dot. Without loss of generality, we assume that the chemical potential of the reservoirs are zero. A non-zero potential energy can be incorporated into $E(v, n)$ via a modification of the chemical potential terms.

   With this, for a fixed electron configuration $n$, we follow [9] and model the sensor peak as

$$\frac{g_{\max}}{k_B T} \cosh^{-2}\left(\frac{E(v, n) - E(v, n \pm e_i)}{k_B T}\right) \ ,$$

where the sign in $E(v, n) - E(v, n \pm e_i)$ is chosen to minimize the absolute value of the energy difference. We will now assume that for fixed $v$ the sensor signal is measured for a fixed time period in which $\ell$ measurements are performed. During this simulated measurement time, the sensor peak is affected by electrostatic noise. In our simulator, we consider two sources of noise: a quickly changing decorrelated noise $\epsilon_1, \ldots, \epsilon_\ell \sim \mathcal{N}(0, \sigma_F^2)$ and a slowly changing correlated noise $\epsilon_S$ that changes much slower than the total measurement time at a single point, i.e., we assume it is constant over the duration of a single measurement. This could be for example the outcome of an Ornstein-Uhlenbeck process. Further, we generalize the model to more than a single electron configuration $n$ by assuming that over the measurement period, the distribution of the mixed state $\rho(v)$ induced by the chosen gate voltages thermalizes and gives the distribution

$$P(n|v) = \frac{1}{\text{Tr}(\rho(v))} \langle n|\rho(v)|n\rangle$$

With this, the sensor peak can be computed by

$$G = \frac{g_{\max}}{k_B T} \sum_n P(n|v) \frac{1}{\ell} \sum_{i=1}^{\ell} \cosh^{-2}\left(\frac{E(v, n) - E(v, n \pm e_i)}{k_B T} + \epsilon_i + \epsilon_S\right) \ .$$

   When $\ell$ is large, computing $G$ can become expensive. Therefore, we use an approximation that has constant computation time in $\ell$. We approximate the distribution of the fast noise

$$\frac{1}{\ell} \sum_{i=1}^{\ell} \cosh^{-2}\left(\frac{E(v, n) - E(v, n \pm e_i)}{k_B T} + \epsilon_i + \epsilon_s\right)$$

as a normal distribution using moment matching. For large $\ell$, the approximation using the normal distribution is valid due to the law of large numbers. Unfortunately, the moments of this distribution have no closed form, so we approximate $\cosh^{-2}(x) \approx 4\phi(x/\sigma_P)$, where $\phi$ is the pdf of the standard normal distribution and $\sigma_P = 1/0.631$ is the width chosen to give the closest match to the peak. In practice, we allow using a much larger value for $\sigma_P$ in order to produce the broad peaks obtained in experiments, where the broadening is likely due to a hybridisation of the sensor peak with the reservoir.

## 4.5   Line-Scans and Optimisations

So far, the simulator assumes that for a given gate-voltage the current electron configuration $n$ is known. This is a difficult assumption, as computing the state minimizing $E(v, n)$ is a

quadratic integer problem, which is NP-hard [17]. We will work around this limitation in the context of 1D line scans where we compute the sensor responses for a set of gate voltages $v_i = v_0 + w \cdot i$ for $i \in 0, \ldots, N_{\text{scan}} - 1$. If the state of the initial set of gate-voltages $v_0$, $n_0$, is known, we can use $P(n_0)$ to check whether $v_1$ is still within the polytope. If it is, we immediately know $n_1 = n_0$. Otherwise, we can compute where on the polytope the line $v_0 + w \cdot t$ intersects with the boundary of the polytope and use the known label to find candidates for the next state $n_1$. This works well in cases where the intersection does not happen close to a vertex, as otherwise $v_1$ might belong to a state that does not share a facet with $P(n_0)$. In that case, we take a different point in $P(n_0)$ and perform a line-scan in direction of $v_1$ through that point and then iterate over the transitions and states along this ramp until state $n_1$ is reached. This process can be repeated for all gate-voltages along the line.

For the initial point, we rely on a good estimate $\hat{n}_0$ by the user. If $v_0$ is not within the polytope, we use the same ramp procedure as described before to find the correct state. The only hard requirement on $\hat{n}_0$ is that it can be attained. This can not happen in the initial simulator, but in simulations of subspaces of the voltage space (called slices) that are, for example generated when the user decides to fix a gate voltage of a compensated sensor dot - In this case, a state might not be attained while keeping the gate voltage of the sensor fixed. For sliced simulations, finding an initial state guess might be challenging, but it is always possible to find a starting point, by computing it on the simulator will full degrees of freedom.

This line scan procedure is still very expensive as computing the set of boundaries for $P(n)$ is a hard problem to compute. To optimise this, we employ a number of techniques. First, we cache all computed polytopes for later use. This is especially useful for 2D scans, where many lines share the same polytopes as well as repeated line scans, e.g., when benchmarking an algorithm that relies on simulated voltage ramps.

Further, we speed up the computation of the boundaries of $P(n)$ by using the heuristic described in the Appendix of [7]. Instead of computing problem (4) using all possible linear inequalities to check whether $v \in P(n)$, we use that any subset of those inequalities leads to a convex polytope $\hat{P}$ such, that $P(n) \subseteq \hat{P}$. Therefore, if an inequality is found not touching $\hat{P}$, it can be removed from the set of transitions without checking any other inequalities. This property allows that for polytopes with a large number of facets, we can create smaller batches of inequalities, filter out irrelevant inequalities and then merge the batches step-by-step in order to only solve problems with a manageable number of inequalities.

## 5  Conclusion

In this work we introduced and presented `QDarts`, a novel simulator for realistic sensor responses and charge stability diagrams. Our simulator is fast and reliable and can scale to medium sized devices of more than 10 quantum dots. We have shown results for a 6 dot system (4 quantum dots + 2 sensor dots), that can simulate a conductance scan on a 100x100 grid in under a minute (on a standard laptop), comparable to typical measurement times on a real device. While it does not include spin physics, it produces results that closely match those of real devices and demonstrates many experimentally observed effects. Our work is complementary to a simultaneously published package called `QDsim`, which leverages a polytope-finding algorithm to efficiently simulate and locate charge transitions in large scale quantum dot arrays [18,19]. `QDarts` also shows parallels to another recently published package called `SimCATS` [20,21], which shares a similar goal.

As an additional goal for `QDarts`, we aimed for delivering a package that efficiently simulates realistic devices, while including an easy-to-use interface. As the name suggests, `QDarts` provides a convenient way of pinpointing specific charge states and their (selected) transitions.

This allows for adapting the simulation to many devices and types of experiment. It also allows for an extension of the constant interaction model that adds realistic scaling behaviour of the charging energies as a function of charge occupation. In conclusion, it presents a significant advancement in our ability to simulate quantum dot arrays. In the future we will work on extending the model further, including spin degrees of freedom and barrier gates.

**Acknowledgements** We acknowledge insightful discussions with Valentina Gualtieri and Eliška Greplová on the construction of a simulator package and interface.

**Funding information** O.K received funding via the Innovation Fund Denmark for the project DIREC (9142-00001B). This work was supported by the Dutch National Growth Fund (NGF), as part of the Quantum Delta NL programme.

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
