# Peer review of "QDarts: A Quantum Dot Array Transition Simulator for finding charge transitions in the presence of finite tunnel couplings, non-constant charging energies and sensor dots"

_SciPost Physics Codebases, doi:SciPost Phys. Codebases 43-r1.0 (2025) , SciPost Phys. Codebases 43 (2025)_

## Round 1 · Referee Report · Matias Urdampilleta (Referee 1) · 2024-7-11

Strengths

1- the present approach and code tackles an important problem in semiconductor spin qubit: how to distribute charges in an array 2- the code is well structured and in comparison with a brute force approach is efficient 3- the code and model allow to track and identify charge state in the array 4- it can reproduce data realistically by accounting for noise model on charge sensor

Weaknesses

1- difficulty to install the code for a non-python expert (no pip install) 2- the code returns some errors for particular points in gate space 3- the model is based on an open system, the possibility to close it and work with a finite number of charge would be a plus

Report

Krzydwa et al. present a physical model of the electrostatic of quantum dots which allows to extract the stable charge configuration of an array of quantum dot and simulate the response of the sensor to charge transition.
This work is very timely as we can see with few papers submitted at the same time on ArXiv which try to adress this need in the community with different approaches .
Moreover, the model is refined to account for the noise and the tunnel coupling and is tested against experimental data which are quantitatively reproduced.
The model is implemented in a python code which is accessible and properly documented. As experimentalists working on this problem, we found the code easy to use (while a bit difficult to install) and quite efficient in term of speed.
We have bench-marked this code against our own which has a more brute force approach and found the present code to be much more efficient in particular toward a large number of quantum dots and charge number. We would therefore encourage experimentalists to use this code to simulate their charge stability diagram of an open array.
There are still some functionalities that are missing and which could definitely benefit to the community to simulate more complex array. These are recommendations, and a bit beyond our expertise to estimate the technical feasibility.
First of all, the ability to work with a finite number of charge in the array would be very helpful to simulate isolated arrays which is a widely used approach in experiments (see Flentje, et al. Nat Commun. 8, 501 (2017) or Meyer et al. Nano Lett. 24, 11593 (2023) or Yang, et al. Nature 580, 350–354 (2020).)
Second, instead of using charge detection, probing the quantum capacitance through gate-based reflectometry would be a nice functionality. For instance, reproducing RF signal on stability diagrams for different parameters such as tunnel coupling, lever arm, frequency etc… would be extremely useful. This readout method is believed to be scalable approach to control and read spins in large array (Crippa, et al. Nat Commun 10, 2776 (2019), Veldhorst, et al. Nat Commun 8, 1766 (2017).)

Requested changes

We do not request particular change apart from facilitating the installation with better documentation.

Recommendation

Publish (meets expectations and criteria for this Journal)

  • validity: top
  • significance: high
  • originality: good
  • clarity: good
  • formatting: good
  • grammar: excellent

Author:  Jan Krzywda  on 2024-10-01  [id 4811]

(in reply to Report 1 by Matias Urdampilleta on 2024-07-11)

I. The model is implemented in a python code which is accessible and properly documented. As experimentalists working on this problem, we found the code easy to use (while a bit difficult to install) and quite efficient in term of speed. We have bench-marked this code against our own which has a more brute force approach and found the present code to be much more efficient in particular toward a large number of quantum dots and charge number. We would therefore encourage experimentalists to use this code to simulate their charge stability diagram of an open array.

Response We appriciate the effort made by the reviewer to test the code and provide feedback. We are glad that the code was found to be efficient and easy to use. At the same time, we are grateful for the suggestion of improving the installation process.

To improve user experience, we have developed transparent documentation and installation instructions, which is available at https://condensedai.github.io/QDarts/. We also provide requirements.txt file to install all necessary packages with one command.

II. There are still some functionalities that are missing and which could definitely benefit to the community to simulate more complex array. These are recommendations, and a bit beyond our expertise to estimate the technical feasibility. First of all, the ability to work with a finite number of charge in the array would be very helpful to simulate isolated arrays which is a widely used approach in experiments (see Flentje, et al. Nat Commun. 8, 501 (2017) or Meyer et al. Nano Lett. 24, 11593 (2023) or Yang, et al. Nature 580, 350–354 (2020).)

Response

We appreciate the suggestion; however, the implementation is challenging in the presence of the sensor dot, which is a necessary element of our readout simulation. The sensor dot has to be connected to the reservoir, while, in principle, the other dots can indeed be decoupled. The presence of an alternative readout, such as in-situ reflectometry measurements from the next point, would allow for writing a wrapper around the CapacitanceModel class to limit the neighbor enumeration of a state. For example, one could call CapacitanceModel.enumerate_neighbours and then remove all returned states that do not conserve the total number of electrons. However, the full implementation of this solution or a partial reservoir connection requires significant effort, which is not aligned with other goals of the project.

In particular, a similar goal can be achieved by implementing transition dynamics into the array, based on barrier gates and tunnel coupling strength. While this approach allows for the implementation of effects like latching, setting the transition speed for certain reservoir transitions to zero can efficiently simulate the effect of fixed charges. We plan to implement this in a follow-up publication.

III. Second, instead of using charge detection, probing the quantum capacitance through gate-based reflectometry would be a nice functionality. For instance, reproducing RF signal on stability diagrams for different parameters such as tunnel coupling, lever arm, frequency etc… would be extremely useful. This readout method is believed to be scalable approach to control and read spins in large array (Crippa, et al. Nat Commun 10, 2776 (2019), Veldhorst, et al. Nat Commun 8, 1766 (2017).)

Response

We are grateful for this suggestion. In order to make the code more versatile and useful for a wider range of applications, we have implemented the vannila version of in-situ reflectometry and provided its description in the repository, providing a separate example file. For the time being we have not included description of this mechanism in the manuscript, as the corresponding user interface is likely to change, once barrier gates are added (work in progress). Nevertheless, below we provide a brief description of the implemented feature from the corresponding notebook example insitu_reflectometry.ipynb:

Following Vigneau et al. (2023), we have implemented the adiabatic version of the quantum capacitance sensing. We use the fact that capacitance is proportional to $\partial P_\mathbf{n}/\partial \epsilon$, where $P_\mathbf{n}$ is the probability of finding charge occupation $\mathbf{n}$ that minimizes electrostatic energy $E_\mathbf{n}$ at given voltage $\mathbf{v}$, and $\epsilon_{\mathbf{n}\mathbf{m}}$ is the energy detuning to the next energy state with charge configuration $\mathbf{m}$. The insitu signal is activated by adding the insitu_axis argument (default: None): ```python

model = CapacitanceModel( ... insitu_axis=[1,0], plane_axes=[[1,0],[0,1]] ... ) ```

where by defining the vector insitu_axis we select which gate is to be modulated. In the example above, the oscilattory signal is coupled to the first axis, which according to insitu_axis is corresponds to the first plunger gate $v_0$. We compute the relative signal strenght using finite difference approximation to the derrivative:

$$ S(\mathbf v) \propto \frac{\partial P_\mathbf{n}}{\partial \epsilon_{\mathbf{n}\mathbf{m}}} \approx \frac{P_\mathbf{n}(\mathbf v + \delta \mathbf v) - P_\mathbf{n}(\mathbf v)}{\epsilon_{\mathbf{n}\mathbf{m}}(\mathbf{v} + \delta \mathbf v)- \epsilon_{\mathbf{n}\mathbf{m}}(\mathbf{v})} $$
where $\delta \mathbf v = \delta v$ (insitu_axis$\cdot$ plane_axes), $e\alpha$ is a constant irrelevant for normalized signal, and we keep $\delta v = 0.01$mV is sufficiently small and fixed.

To compute the signal $S(\mathbf{v})$ at every point in voltage space we contruct two-level Hamiltonian, using the subspace of two lowest lying energy states,

$$ H_{2\text{-level}}(\mathbf{v}) = \begin{pmatrix} E(\mathbf{n},\mathbf{v}) & t_{\mathbf{n}\mathbf{m}}(\mathbf{v}) \ t_{\mathbf{n}\mathbf{m}}(\mathbf{v}) & E(\mathbf{m},\mathbf{v}) \end{pmatrix} $$
where $E(\mathbf{n},\mathbf{v})$ and $E(\mathbf{m},\mathbf{v})$ are the ground and excited state electrostatic energies corresponding to charge occupation $\mathbf{n}$ and $\mathbf{m}$, and $t_{\mathbf{n}\mathbf{m}}$ is the tunnel coupling between them, which is either constant or zero if two states are not coupled. Using the Hamiltonian one can compute relevant quantities, such as the energy detuning $\epsilon_{\mathbf{n}\mathbf{m}}(\mathbf{v})=E(\mathbf{n},\mathbf{v})-E(\mathbf{m},\mathbf{v})$ and the probability of being in on of the charge configurations
$$ P_\mathbf{n}(\mathbf{v}) = \frac{1}{2} \left(1 - \frac{\epsilon_{\mathbf{n}\mathbf{m}}(\mathbf{v})}{\sqrt{\epsilon_{\mathbf{n}\mathbf{m}}(\mathbf{v})^2 + 4t_{\mathbf{n}\mathbf{m}}^2(\mathbf{v})}}\right) $$
Intuitively we assume the signal is proportional to the change in quantum capacitance, caused by motion of the electron between the dots induced by the modulation of the gate voltages. At this point, we have not included the other effects contributing to in-situ signal including sisyphus resistance and tunneling capacitance, however their contribution is expected to be relatively weaker [Vigneau et al. (2023)].

Below we are attaching the notebook realising current version of the insitu reflectometry code, for the example of the two dots system: https://github.com/condensedAI/QDarts/blob/v0.1/examples/insitu_reflectometry_beta.ipynb

We are attaching the differential file, with the marked changes

Attachment:

QDarts_diff_file_bMZwNyz.pdf

---

## Round 1 · Referee Report · Anonymous (Referee 2) · 2024-8-5

Strengths

1- addresses a timely problem 2- package functionalities go beyond the minimal electrostatic model, e.g., by incorporating the effects tunnel coupling and sensor noise 3- nicely written paper

Weaknesses

1- admittedly, there are important functionalities missing, e.g., spin effects, orbital level spacing 2- the even-odd tunnel coupling effect is not incorporated (see report for details) 3- a more detailed comparison with similar packages would be welcome

Report

The work by Krzywda et al. presents a python package to simulate charge stability diagrams of capacitively- and tunnel-coupled quantum dot systems. The user can specify the parameters of the device (capacitance network parameters, noise characteristics of the sensing dot, etc.), and the code outputs the charge stability diagram.

I know that these types of simulations are routinely done in many experimental and theoretical research groups worldwide (as well as the few companies building quantum hardware based on such devices) using in-house-built software solutions. Therefore I see the present package as a welcome attempt to provide a general tool for this community, which is likely to make research and engineering in the field more efficient.

The package has a number of functionalities that go beyond the minimal model of electrostatics with the constant-interaction model: e.g., gate virtualisation, charge-dependent capacitances, noise on the sensor dots, etc. Admittedly, there are also important functionalities that are often needed but missing from the present implementation: e.g., spin degree of freedom, orbital level spacing, role of the barrier gates, etc.

The authors benchmark their package against a selection of state-of-the-art experiments, which I find mostly satisfactory.

(i) There is one point, however, where I am uncertain if the model used here is appropriate. This has to do with the simulation of the finite tunnel coupling effect, shown, e.g., in Fig. 8. My understanding is that in reality (more precisely, on the level of the Hubbard model), there is an even-odd effect for the tunnel-coupling-induced features of the charge stability diagram. For example, the tunnel coupling between the (1,0) and (0,1) charge configurations is weaker than the tunnel coupling between the (1,1) and (0,2) charge configurations by a factor of sqrt(2). In turn, this difference should be visible in any charge stability diagram that includes both transitions. However, the tunneling model specified by the authors, see Eq. (1), does not include this effect. My feeling is that this effect could be easily incorporated in the model, the only price to be payed is a slightly more elaborate formula instead of Eq. (1). I request the authors to implement this change, or if they decide not to do so, then at least comment on this issue in the revised version of the manuscript.

(I do understand that this even-odd effect is related to the spin degree of freedom, which is admittedly missing from the current version of the package, but I also feel that this effect is qualitatively different from other spin effects, and also straightforward to implement in the existing framework. )

(ii) In the Conclusion, the authors list QDsim and SimCATS as software tools that are similar to QDarts. From the viewpoint of a potential user who wants to pick one of these potential solutions, I find it desirable to have a more detailed comparison of these tools, describing similarities and differences. Ideally, this could come in the form of a table comparing functionalities and performance, but if that is too much to ask for, then anything going beyond the current description would be welcome.

(iii) My final request is to add color scales and physical units to the conductance plots (Fig. 1, Fig. 7, Fig. 8). Perhaps one could trace these back from the example python notebooks; still I feel this is important, to make the paper self-contained and complying to the principles of accuracy and reproducibility.

Requested changes

Please see (i), (ii), and (iii) in my report.

Recommendation

Publish (meets expectations and criteria for this Journal)

  • validity: high
  • significance: high
  • originality: good
  • clarity: high
  • formatting: good
  • grammar: excellent

Author:  Jan Krzywda  on 2024-10-01  [id 4810]

(in reply to Report 3 on 2024-08-05)
Category:
remark
answer to question
validation or rederivation

I There is one point, however, where I am uncertain if the model used here is appropriate. This has to do with the simulation of the finite tunnel coupling effect, shown, e.g., in Fig. 8. My understanding is that in reality (more precisely, on the level of the Hubbard model), there is an even-odd effect for the tunnel-coupling-induced features of the charge stability diagram. For example, the tunnel coupling between the (1,0) and (0,1) charge configurations is weaker than the tunnel coupling between the (1,1) and (0,2) charge configurations by a factor of sqrt(2). In turn, this difference should be visible in any charge stability diagram that includes both transitions. However, the tunneling model specified by the authors, see Eq. (1), does not include this effect. My feeling is that this effect could be easily incorporated in the model, the only price to be payed is a slightly more elaborate formula instead of Eq. (1). I request the authors to implement this change, or if they decide not to do so, then at least comment on this issue in the revised version of the manuscript. (I do understand that this even-odd effect is related to the spin degree of freedom, which is admittedly missing from the current version of the package, but I also feel that this effect is qualitatively different from other spin effects, and also straightforward to implement in the existing framework. )

Response We appreciate the suggestion. We have implemented this observation into the simulator and included it also in the manuscript. To be precise, when the number of electrons on the pair of dots affected by a transition is even, we increase the tunnel coupling by sqrt(2). We included the change in the manuscript

In practice, the value of the effective tunnel coupling is influenced by the parity of the charges involved in the transition. Specifically, if the number of electrons on the pair of quantum dots affected by the transition is even, the tunnel coupling increases by a factor of (\sqrt{2}) \cite{vanVeenPRB2019}. While description of this effect would require considering the spin degree of freedom, it has been already implemented in the simulator due to its simplicity. This means that $t_{ij}$ in Eq. (1) is multiplied by (\sqrt{2}), if the number of electrons on the pair of dots is even, for instance $(2n-k,k) \to (2n-k-1,k+1)$.

II In the Conclusion, the authors list QDsim and SimCATS as software tools that are similar to QDarts. From the viewpoint of a potential user who wants to pick one of these potential solutions, I find it desirable to have a more detailed comparison of these tools, describing similarities and differences. Ideally, this could come in the form of a table comparing functionalities and performance, but if that is too much to ask for, then anything going beyond the current description would be welcome.

Response We agree with the reviewer that transparency in comparison between different packages is highly beneficial for the community and can possibly lead to their convergence into one fully-functional and widely used simulation package. Instead of table, which for continously evolving software might provide too restrictive information that quickly outdate itself, we have included additional description of each package, indcluding the forth one. We decided to hint at the philosophy against each of the packages and provide the most typical use cases.

Our work complements three simultaneously published packages. To understand the relationships between them, we will explain their main ideas and propose the most suitable use cases. The package QDsim leverages direct solvers of the integer optimisation problem to efficiently simulate and locate charge ground states in large-scale quantum dot arrays. It is suitable for simulating the electrostatic global properties of large arrays (up to 100 dots), where thermal effects, sensor signals, and tunnel coupling effects are less relevant. It can also derive approximate device parameters based on the device model, making it useful during the design stage. Similarly, the QArray package focuses on large devices and implements a constant interaction model. It computes the expected number of charges at a fixed temperature. Crucially, it is characterised by high-performance computation achieved through its implementation in the Rust programming language and the specialised computing library JAX. As a result, this package is mostly suitable for data-driven applications, where a fast rate of data generation is more important than capturing detailed quantum effects. Finally, the SimCATS package provides a comprehensive tool for fast simulations of double quantum dot transitions. It includes tools for modelling charge transitions and the influence of tunnel coupling through Bézier curves. It also allows for implementing distortions that reflect experimental reality, such as noise and finite temperature. It is suitable for simulating charge stability diagrams that include sweeps of two voltage gates, where the polytope-finding algorithm and the identification of charge transitions for arbitrary voltage combinations are less relevant. From this perspective, the QDarts package aims to provide fast and faithful simulations of moderate-sized quantum dot devices. As the name suggests, it offers a convenient way of pinpointing specific charge states and selecting transitions in arbitrary cuts through voltage space. QDarts simulates tunnel coupling effects and includes a physical model of the sensor dot, which incorporates a correlated noise model. Additionally, it extends the constant interaction model by adding the scaling behaviour of the charging energies as a function of charge occupation. Together, this makes QDarts a suitable package for the autonomous tuning of quantum dot devices, where reconstructing experimentally observed deviations from the constant interaction model may become important. In conclusion, QDarts represents a significant advancement toward fast and reliable simulations of moderate-sized quantum dot arrays. Future versions of the package are planned to include spin degrees of freedom, barrier gates, and experimentally relevant effects such as latching and in-situ capacitance sensing.

III My final request is to add color scales and physical units to the conductance plots (Fig. 1, Fig. 7, Fig. 8). Perhaps one could trace these back from the example python notebooks; still I feel this is important, to make the paper self-contained and complying to the principles of accuracy and reproducibility.

Response

Appreciating the feedback, we have provided the colorbar in both the manuscript and the example file, including the colorscale in every figure where it applies. Due to the qualitative nature of the charge stability diagram and the complexity of the experimental setup, we decided to work with normalized conductance and capacitance values, i.e., values divided by the maximum value of the respective quantity. A similar strategy is often employed in the literature, even for experimental data, where either normalized conductance (D. Schröer et al, PRB 2007) or "a.u." units (N. W. Hendrix et al., Nature 2020) are used.

We are attaching the diff file with changes with respect to original file

Attachment:

QDarts_diff_file.pdf

---

## Round 3 · Author Response

After addressing the issues raised by the reviewers, we have submitted a revised version of the manuscript and code, which includes new functionalities. We are grateful for the review process, as it has helped us achieve these improvements and provided valuable guidance for further development of the code.

---

## Round 3 · List of Changes

Most of the changes address the reviewers' suggestions (see response). The key changes are as follows:

Manuscript: - A more detailed comparison between QDarts and other similar packages. - Added colorbars to indicate parameters in the figures. - Explanation of the odd-even parity effect for tunnel coupling. Code: - Implementation of the odd-even parity effect for tunnel coupling. - Measurement of quantum capacitance via in-situ reflectometry (new example notebook). - A new example notebook for the low-level interface. - Updated documentation, including an improved installation guide and requirements. - Overall improvements to the code structure and quality.

---

## Editorial Decision

published